# The Isolation and Characterization of Perlucin in Pacific Abalone, *Haliotis discus hannai*: A Shell Morphogenic Protein with Potential Responses to Thermal Stress and Starvation

**DOI:** 10.3390/biology13110944

**Published:** 2024-11-18

**Authors:** Yusin Cho, Md Abu Hanif, Shaharior Hossen, Soo Cheol Kim, Ji Do Han, Doo Hyun Cho, Kang Hee Kho

**Affiliations:** 1Department of Fisheries Science, Chonnam National University, Yeosu 59626, Republic of Korea; joyusin693@gmail.com (Y.C.); mhanif713@jnu.ac.kr (M.A.H.); shaharior.pstu@gmail.com (S.H.); kijin23471@gmail.com (D.H.C.); 2South Sea Fisheries Research Institute, National Institute of Fisheries Science, Yeosu 59780, Republic of Korea; sckim9194@korea.kr (S.C.K.); onemap83@gmail.com (J.D.H.)

**Keywords:** secreted protein, shell biomineralization, carbohydrate binding, starvation, thermal stress

## Abstract

Mollusk mantle tissue secretes organic matter and inorganic ions to form a rigid shell that covers and protects their soft bodies from predators and environmental factors. Among the various organic molecules, proteins are of great importance for mollusk shell crystallization. Although several shell matrix proteins have been identified, detailed research on these proteins remains scarce. Therefore, this study aimed to isolate the shell biomineralizing matrix proteins and investigate their roles in Pacific abalone, *Haliotis discus hannai*. In the present study, a shell matrix protein, perlucin (denoted as *Hdh-Perlucin*), was identified from Pacific abalone for the first time through molecular cloning; it was characterized and its expression pattern was observed to reveal its function. Sequence analysis revealed that this protein possesses a signal peptide, a C-type lectin domain, and a signature motif within its mature peptide, exhibiting carbohydrate-binding activity. *Hdh-Perlucin* was significantly expressed in the mantle tissue of adult Pacific abalone, during larval development (shell formation stages), in rapidly growing juveniles, and during injured-shell remodeling, indicating its role in shell biomineralization. However, thermal stress and starvation were found to influence the expression pattern of *Hdh-Perlucin*. These findings may be useful for future advanced molecular research in abalone and other mollusks.

## 1. Introduction

In nature, a myriad of living organisms synthesize biominerals for mineral ion storage, tissues, soft-body support, protection from predators, and environmental factors [1]. Biomineralization in mollusks offers a wide range of microstructures and evolutionary origins [2]. Furthermore, after coral mineralization, the secretion of shells by mollusks is likely one of the most prevalent and widely occurring biomineralization processes in the metazoan realm [3]. The growth of molluscan shell crystals is predominantly thought to be initiated from solution by an extracellular organic matrix. The molluscan shell is an organo-mineral biocomposite, of which inorganic minerals (CaCO_3_) make up 95–99% while organic matter constitutes 0.1–5% [4]. During molluscan shell formation, the mantle tissue secretes organic matter (carbohydrates, lipids, proteins, glycoproteins, peptides, chitin, and polysaccharides), and inorganic ions are transported to the extrapallial space and deposited as mineralized structures [5]. These interactions result in the formation of microstructures made of calcite and aragonite. Despite the recent significant progress in biomineralization on the formation, composition, and patterning of molluscan shells, ongoing debates remain regarding the mechanisms involved [6].

Currently, more than 100,000 living mollusk species from aquatic and terrestrial environments have been reported to secrete shells [7]. The initial molluscan shell formation process starts at the post-gastrula stage [8], whereby the ectodermal cells thicken to form the shell field [9]. Then, the shell field folds inward into the blastocoel (glandular canal/shell plate), developing into the shell gland during the early trochophore stage. The shell field expands through mitotic divisions and cell flattening to form the calcifying mantle [3]. The first shell, protoconch-I (prodissoconch 1 in bivalves [10]), becomes visible at the end of the trochophore stage within 18 to 25 h of fertilization, while protoconch-II (prodissoconch 1 in bivalves) is formed during the veliger larval stage [11]. Subsequently, the adult shell develops after the veliger larvae settle and metamorphose into juveniles. In bivalves, the onset of adulthood is evidenced by the formation of foot tissue in the pediveliger stage and a marked transition in the micro-texture between the existing larval shell and the newly formed adult dissoconch [12]. Gene expression studies have revealed several conserved transcription factors and signaling ligand genes expressed in discrete zones within and around the developing shell field [9,13].

Insights into biomineralization have identified that fibronectin type III (FN3), epidermal growth factor (EGF), chitin binding (CB), and whey acidic protein (WAP) all play key roles in shell formation [2,14,15,16,17,18]. The mineralized shells usually possess several layers with morphological and textural diversity. Previously, several shell biomineralizing genes including shell matrix protein 5 [5], secreted protein acidic and rich in cysteine [19], carbonic anhydrase I [20], and carbonic anhydrase II [21] have been identified and characterized in Pacific abalone. Additionally, genotyping by sequencing in Pacific abalone identified some single-nucleotide variants of monocarboxylate transporter 12, chitin-binding domain-containing protein, zinc finger protein 850-like, and carbonic anhydrase, which had previously been reported to associate with shell formation in mollusks [22]. A variety of shell matrix proteins (SMPs) responsible for the inner layer (nacreous layer) formation in mollusks have been identified and studied, including nacrein, lustrin A, perlustrin, and perlucin, although their involvement in shell-building processes remains largely unknown.

Perlucin, a shell matrix protein, was originally isolated from the inner shell (nacreous layer) of abalone [15,23]. The perlucin amino acid sequence exhibits a high similarity to the carbohydrate-recognition domain (CRD) or C-type lectin domain (CTLD) and other members containing C-type lectins [24]. Perlucin induces the precipitation of CaCO_3_ from the highly impregnated solution, suggesting that it may facilitate the nucleation and crystal growth of CaCO_3_. Therefore, perlucin is a constitutive nacre protein and an essential molecule in regulating shell and pearl formation [25]. To date, the role of perlucin in shell biomineralization has been reported from a variety of invertebrate species, including freshwater mussel, *Hyriopsis cumingii*; greenlip abalone, *Haliotis laevigata*; eastern oyster, *Crassostrea virginica*; blue mussel, *Mytilus edulis*; Pacific oyster, *Crassostrea gigas*; variously colored abalone, *Haliotis diversicolor*; disk abalone, *Haliotis discus discus*; pearl oyster, *Pinctada fucata*; and whiteleg shrimp, *Litopenaeus vannamei*.

The abalone shell has been a key model for exploring the biomineralization mechanisms in gastropod shells. Meanwhile, Pacific abalone is the most popular seafood and widely cultured marine gastropod mollusk in South Korea, China, and Japan [19,26]. In nature, abalones are often exposed to various stressors, including thermal stress (which causes summer mortality), nutritional stress, and injury from hard particles or predation. Although shell matrix proteins make up only a small part of the shell, they are the main macromolecules responsible for regulating biocrystal formation [27] and can protect abalone from these stressors through activation and functionality. Perlucin is one such shell matrix protein involved in biomineralization [28]. However, while perlucin has been identified in some molluscan species, the isolation and characterization of this shell matrix protein still needs to be performed in Pacific abalone. Therefore, this study aimed to isolate and characterize the perlucin gene from Pacific abalone to identify its involvement in shell formation and response to thermal stress and starvation.

## 2. Materials and Methods

### 2.1. Experimental Animals and Tissue Collection for Isolation of the Perlucin Gene

For mRNA isolation and expression analysis, three individual Pacific abalone (two years old), with an average body weight of 27.6 ± 0.54 g and an average total length of 14.08 ± 0.39 cm, were obtained from a shrimp farm in Jindo-gun, South Korea. After applying 5% MgCl_2_ anesthesia, tissue samples were collected from the cerebral ganglion (CG), muscle (MUS), mantle (MNT), gill (GIL), testis (TES), ovary (OVR), hemocyte (HEM), and digestive gland (DG). These tissue samples were cleaned with PBS, rapidly flash-frozen in liquid nitrogen (LN2), and stored for total cellular RNA extraction at –80 °C.

### 2.2. Samples of Early Development Stages

Thirty reproductively mature (three years old) Pacific abalone (20 females and 10 males) were induced to spawn for artificial fertilization through air exposure and UV-treated water. Gametes were successfully released from three females and two males. Artificial fertilization was carried out following spawning, as previously described by Hanif et al. [19]. After fertilization, samples were collected at various developmental stages: fertilized egg (FE), 2-cell (2-CL) and 8-cell (8-CL) stages, morula (MOR), blastula (BLS), trochophore (TRP), veliger (VLG), and juvenile (JUV). These samples were promptly flash-frozen in LN2 and stored at –80 °C until the extraction of total cellular RNA.

### 2.3. Tissue Sample Collection from Different Growth Types of Pacific Abalone

After culturing in a tank for one year, the abalone were collected and sorted according to their size. Three individuals from each category (slow growth, average growth, and rapid growth) were anesthetized using 5% MgCl_2_, and muscle tissue was collected to determine the expression profiles of the different Pacific abalone growth types. Tissue samples were rinsed with PBS (0.1 M), rapidly flash-frozen in LN2, and stored at –80 °C until total RNA extraction. The average size for slow growth was 4.6 ± 0.2 cm, for average growth, it was 7.2 ± 0.5 cm, and for rapid growth, 10.7 ± 0.3 cm.

### 2.4. Collection of Tissues During Biomineralization of Damaged Shells

Juvenile Pacific abalone (ten months old) were used for the shell biomineralization experiment. Thirty juveniles were split into two groups: the first group (group 1) acted as the control, while the second group (group 2) received shell damage treatment. In group 2, a perforator was used to puncture the shell of each abalone near the mantle tissue with a 1 mm round orifice. The abalones were subsequently reared in a tank with constant aeration, water flow, and regular feeding. The water temperature in the tanks was maintained at 19.2 ± 1.8 °C throughout the experiment. Mantle tissues were collected weekly from five individuals in each group, while tissues from control abalone were sampled at the start of the experiment. The collected tissues were rinsed with PBS, flash-frozen in LN2, and stored at –80 °C until total RNA extraction.

### 2.5. Collection of Tissue Sample from Pacific Abalone Under Starved Conditions

Mature Pacific abalones (aged two years) were collected from sea cages in Jindo-gun and moved to an abalone hatchery in Yeosu, South Korea. The collected abalones were kept in tanks with seawater and adequate food for 15 days to acclimate. Subsequently, the acclimatized abalones (*n* = 48) were split into two groups: group A was the control group, and group B was subjected to starvation. After anesthesia, as previously described, three abalones from each group were sacrificed weekly for three consecutive weeks. Mantle tissues were collected from groups A and B during this period, as well as on the day following re-feeding. Samples were then stored at –80 °C for total cellular RNA extraction using the same protocol.

### 2.6. Collection of Tissue Samples from Pacific Abalone Exposed to Thermal Stress

To explore the impact of thermal stress on *Hdh-Perlucin* expression, abalone were subjected to heat treatments at 15 °C, 25 °C, and 30 °C. Pacific abalones, with an average shell length of 86.2 ± 0.39 mm and a weight of 122.27 ± 0.31 g, were collected from sea cage culture areas and transferred to an abalone hatchery, in which they were reared for approximately 20 days with an ample supply of food and water for acclimation, during which the water temperature was consistently maintained at 20.4 °C. Afterward, they were placed in three separate aquariums for 10 h to acclimate to the temperature-controlled seawater. The first aquarium’s temperature was maintained at 15 °C. In comparison, after 10 h, the temperature increased by 1 °C every two hours and 1 °C per hour to reach 25 °C and 30 °C in the second and third aquariums, respectively. Once the target temperature was achieved, mantle tissue was collected from three individuals in each treatment group, as well as from the control abalones, at 1, 6, 12, 24, and 48 h. The tissues were rinsed with PBS, snap-frozen in LN2, and stored immediately at –80 °C until the extraction of total cellular RNA.

### 2.7. Cloning of the Hdh-Perlucin cDNA Sequence from the Pacific Abalone

RT-PCR was performed to clone the fragment sequence of *Hdh-Perlucin* using cDNA from the mantle tissue, along with specific forward and reverse primers for the target gene and GoTaq^®^ DNA polymerase (Promega, Madison, WI, USA). The primers (forward and reverse) used in fragment sequence cloning were designed based on the perlucin mRNA sequence of *H. diversicolor* (accession No. GU446716.1). Following the fragment sequence cloning, RACE-PCR was carried out using a SMARTer^®^ RACE 5/3 kit (Takara Bio, Shiga, Japan) to acquire the complete sequence of perlucin cDNA. RACE-PCR was performed as described by Hanif et al. [19]. Finally, the RT-PCR and RACE-PCR sequences were combined to obtain the full-length sequence. The primers utilized in this study are listed in Table 1.

### 2.8. Analysis of Hdh-Perlucin Protein

Various online bioinformatics tools were employed to analyze the nucleotide and protein sequences of the cloned Hdh-Perlucin sequence. The protein-encoding segment was analyzed from the full-length cDNA sequence using the ORF Finder online tool. The gene structure was determined using the Scipio eukaryotic gene identification tool (https://www.webscipio.org, accessed on 28 August 2024). The protein identity and similarity (homology) of Hdh-Perlucin was computed through the Sequence Identity and Similarity (SIAS) online tool (http://imed.med.ucm.es/Tools/sias.html, accessed on 28 August 2024). The isoelectric point (pI) and molecular mass of the protein were determined using the ProtParam online tool (https://web.expasy.org/protparam, accessed on 6 September 2024). The perlucin sequences of different mollusks were aligned using the Clustal Omega online tool (https://www.ebi.ac.uk/Tools/msa/clustalo/, accessed on 1 September 2024). The protein sequence alignment was edited and visualized using Jalview software (version 2.11.1.7). The conserved domains of the isolated protein were identified using the SMART online tool (http://smart.embl-heidelberg.de/, accessed on 5 September 2024) and the conserved domain search program provided by NCBI. The Gene Ontology for Hdh-Perlucin was predicted through the protein structure and function prediction website at https://zhanggroup.org/C-I-TASSER/ (accessed on 3 September 2024).

### 2.9. Phylogenetic Analysis

The amino acid sequences of the perlucin protein were retrieved from the NCBI database and aligned using Clustal Omega to create a phylogenetic tree. The amino acid sequences used for the construction of phylogenetic trees has been presented in the Appendix A. A molecular phylogenetic tree was constructed to clarify evolutionary relationships by applying the aligned data through the maximum-likelihood method in MEGA software (version 11). Subsequently, the molecular phylogenetic tree was edited and further visualized using iTOL software (https://itol.embl.de/, accessed on 3 September 2024).

### 2.10. Three-Dimensional Modeling of Hdh-Perlucin

The 3D structure of the Hdh-Perlucin protein was obtained using the C-I-TASSER online protein structure prediction website (https://zhanggroup.org/C-I-TASSER/, accessed on 3 September 2024). The predicted structure was subjected to validation checks using an online protein structure validation website at https://saves.mbi.ucla.edu/ (accessed on 8 September 2024) and further refined using the GalaxyWEB online server at https://galaxy.seoklab.org/ (accessed on 8 September 2024). Finally, the protein structure was visualized using PyMOL (version 3.0.5).

### 2.11. Expression Analysis Using qRT-PCR

Quantification of the *Hdh-Perlucin* mRNA expression levels in Pacific abalone was performed using various tissues and muscle tissues from different growth types. Gene expression analysis was carried out as described by Hanif et al. [19]. A 20 μL reaction mixture was prepared for qRT-PCR, comprising 1 μL cDNA template, 10 μL SyGreen Mix, 1 μL each of primers (forward and reverse), and 10 μL ultrapure water. Three replicates were performed for expression analysis of Hdh-Perlucin, and an internal reference gene (*Hdh-β-Actin*) was used for each sample. The amplification PCR conditions were set following Hanif et al. [19]. After completion, the relative mRNA expression levels were calculated as described by Hanif et al. [19]; the Pacific abalone β-actin gene was used as a reference gene. Details of all the primers utilized in the qRT-PCR analysis are provided in Table 1.

### 2.12. Statistical Analysis

All the mRNA expression values were analyzed and presented as means with standard errors. Variations in the mRNA expression levels were assessed using ANOVA in GraphPad Prism 8.0.1 software. Statistical significance was set at *p* < 0.05 and *p* < 0.001.

## 3. Results

### 3.1. Pacific Abalone Perlucin Sequence

The full-length cDNA sequence of the *H. discus hannai* perlucin gene (*Hdh-Perlucin*) (GenBank accession No. OM937905.1) was cloned from the mantle tissue of Pacific white shrimp. The complete cDNA sequence was 1002 bp long, revealing a 492 bp ORF that encodes a protein (protein ID: UTD53617.1) consisting of 163 deduced amino acid residues, along with a 71 bp 5′-untranslated region (UTR) and a 439 bp 3′-UTR (Figure 1). The calculated molecular weight of the Hdh-Perlucin amino acid residues was 18.4128 kDa, with an isoelectric point of 4.84.

### 3.2. Main Features of the Hdh-Perlucin Amino Acid Sequence

The Hdh-Perlucin protein sequence contains a carbohydrate-binding C-type lectin domain (CTLD) spanning amino acid positions 23C to 151E. The CTLD signature motifs are located within this region at positions 127C to 150C. The sequence features two N-myristoylation sites at positions 16G to 21Q and 85G to 90G. A cAMP- and cGMP-dependent protein kinase phosphorylation site was also found at positions 131K to 134S. Two CK2 phosphorylation sites are located in the amino acid sequence at positions 43S to 46E and 63T to 66E, along with a PKC phosphorylation site that exhibits an [ST]-X-[RK] structure from 80T to 82K. Furthermore, a chitin-binding type-2 domain profile was detected between positions 124S and 150C.

While numerous residues were unconserved, the amino-terminal region exhibited comparatively higher conservation than the carboxyl-terminal region. Among the analyzed amino acid sequences, no signal peptide was detected in *Mytilus trossulus* and *Pinctada fucata*. However, the signal peptides in the remaining species, including three *Haliotis* species, were identical (Figure 2). Six cysteine residues were conserved across the perlucin sequences analyzed in multiple sequence alignments. Interestingly, Hdh-Perlucin contains an additional seventh cysteine residue (Figure 1).

### 3.3. Homology Analysis

The amino acid-based homology analysis revealed that the Hdh-Perlucin sequence from Pacific abalone shared the highest identity (82.82%) with Hd-perlucin from the variously colored abalone (Table 2). Despite belonging to the same genus, Hdd-perlucin from disk abalone exhibited only 39.26% identity with Hdh-Perlucin. Among the mollusk species analyzed, the triangle shell mussel (*Hyriopsis cumingii*, Hc-perlucin) had the lowest identity (36.02%). The least similar identity was observed in the Solanum fruit fly (*Bactrocera latifrons*, Bl-perlucin), among all the species included in the homology analysis.

### 3.4. Structure of Hdh-Perlucin Protein

The tertiary structure of Hdh-Perlucin is composed of 55.83% coils, 23.31% strands, and 20.86% helices. The amino acid sequence includes three helices and eight strands. Notably, all helices are located in the N-terminal region, whereas the C-terminal region is predominantly composed of coils and strands. Additionally, a mannose ligand was identified within the protein structure (Figure 3). The predicted mannose-binding sites are located at residues 117E (glutamic acid), 119N (asparagine), 131K (lysine), 138D (aspartic acid), 139D (aspartic acid), and 140N (asparagine).

### 3.5. Gene Ontology (GO) Analysis

GO analysis predicted that Hdh-Perlucin is an extracellular membrane protein. Furthermore, the principal molecular function of Hdh-Perlucin is binding to carbohydrates (confidence score of 0.66) and carbohydrate derivatives (confidence score of 0.52; Appendix A). The types of carbohydrates could include monosaccharides, such as mannose, while the carbohydrate derivatives may consist of glycosaminoglycans.

### 3.6. Phylogenetic Relationship

A phylogenetic tree was constructed to elucidate the evolutionary relationships between Hdh-Perlucin from Pacific abalone and perlucin proteins from other invertebrates, including mollusks and arthropods. The analysis revealed that mollusk and arthropod perlucin proteins formed distinct clusters (Figure 4). Notably, *Haliotis* perlucin 2 grouped separately with two arthropod species, *Folsomia candida* and *Orchesella cincta*. The mollusk cluster primarily consisted of aquatic gastropods and bivalves, whereas the arthropod cluster was dominated by terrestrial insects, with some crustaceans, including shrimp and crabs, also present. Within the mollusk group, Hdh-Perlucin clustered closely with Hd-perlucin and was phylogenetically closest to Hdd-perlucin 8.

### 3.7. Tissue-Specific Expression of Hdh-Perlucin in Pacific Abalone

Expression analysis across different tissues in Pacific abalone revealed that *Hdh-Perlucin* mRNA was significantly expressed in the mantle (MNT), hemocytes (HEMs), gills (GILs), and digestive gland (DG) (*p* < 0.05; Figure 5). In contrast, significantly lower expression levels were observed in the ovary (OVR) and cerebral ganglion (CG) (*p* < 0.05). While some expressions were detected in the muscle (MUS) and testis (TES), these levels were notably lower compared with those in MNT, HEM, GIL, and DG.

### 3.8. Hdh-Perlucin mRNA Expression During Early Development Stages of Pacific Abalone

Differential *Hdh-Perlucin* mRNA expressions were observed across the early developmental stages. Minimal expression was detected from the fertilized egg (FE) to the morula (MOR) stage, with no significant differences. Expression slightly increased during the blastula (BLS) stage compared with earlier stages (FE, 2-CL, 8-CL, and MOR). A significant rise (*p* < 0.05) was noted after the blastula stage, particularly in the trochophore (TRP) and veliger (VLG) stages (Figure 6). The highest expression level was recorded in juvenile abalone (JUV stage).

### 3.9. Hdh-Perlucin Expression in Pacific Abalone of Different Growth Types

The *Hdh-Perlucin* mRNA expression was significantly upregulated (*p* < 0.05) in rapid-growth individuals compared with that in slow-growth ones (Figure 7).

### 3.10. Hdh-Perlucin mRNA Expression During Shell Regeneration in Pacific Abalone

Upregulated *Hdh-Perlucin* mRNA expression was observed during the shell biomineralization process in Pacific abalone. In the first week following shell damage, expression levels were slightly lower than in the control, though not significantly. However, a substantial increase in relative mRNA expression was observed during the subsequent weeks, surpassing the control levels for the remainder of the experimental period (Figure 8).

### 3.11. Hdh-Perlucin mRNA Expression in Heat-Stressed Pacific Abalone

Differential *Hdh-Perlucin* mRNA expressions were observed during thermal stress, depending on the temperature. After 1 h of heat-stress treatment at 15 °C and 30 °C, the expression levels were similar to that in the control; however, levels were slightly higher at 25 °C. Comparatively, after 6 h, *Hdh-Perlucin* expression significantly increased at 30 °C, while 15 °C and 25 °C showed upward trends, but these remained non-significant. After 12 h, expression significantly increased at 15 °C and 25 °C, whereas a drastic reduction was observed at 30 °C, even falling below the control levels. After 24 h of heat stress, expression continued to increase at 15 °C but decreased at 25 °C and 30 °C. At this point, expression at 25 °C was similar to the levels observed at 6 h, while at 30 °C it was close to zero. After 48 h of thermal stress, expression at 15 °C and 30 °C remained unchanged but decreased at 25 °C, falling slightly below the control levels (Figure 9).

### 3.12. Hdh-Perlucin Expression During the Long Starvation Period

Significant differences in *Hdh-Perlucin* expression were observed in the mantle tissues under starvation-induced nutritional stress. The expression levels were downregulated throughout the starvation period, with a sharp decline noted in the third (3W) and fourth (4W) weeks (Figure 10). However, following re-feeding after prolonged starvation, *Hdh-Perlucin* expression significantly increased, reaching nearly twice the levels observed in the control group.

## 4. Discussion

Perlucin, a shell nacre protein, has been shown to enhance the crystallization of CaCO₃ and to modify its crystal morphology [29]. To our knowledge, this study represents the first report on the isolation and characterization of the perlucin gene from Pacific abalone. The full-length cDNA is 1002 bp, encoding a protein of 163 amino acids, with a 20 bp signal peptide at its N-terminal region. The presence of this signal peptide suggests that Hdh-Perlucin is an extracellular secreted protein [30], containing six cysteine residues that may form disulfide bonds, a feature common to other members of this protein superfamily [31]. The main characteristic of the Hdh-Perlucin amino acid sequence is the presence of a Ca²⁺-dependent carbohydrate-binding C-type lectin domain (CTLD) and a chitin-binding type 2 domain. Lectins (secreted fluid proteins) from Pinctada penguin are known to participate in shell biomineralization [32]. Indeed, chitin and shell matrix proteins (SMPs) play a crucial role in mollusk shell biomineralization [33]. Chitin synthases are reported to be expressed in the outer mantle epithelium, where they contribute to the framework necessary for shell calcification [34]. The presence of an EPN motif suggests that Hdh-Perlucin is a mannose-binding protein [35], and the tertiary structure of Hdh-Perlucin also possesses a mannose ligand. Additionally, GO analysis predicted the molecular function of Hdh-Perlucin to be carbohydrate binding, with specificity for mannose (Appendix A). Similarly, the presence of a C-type lectin domain with mannose specificity was reported in a previous study on the nacre protein in *H. laevigata* [24,25]. Mannose, a monosaccharide in the aldohexose series of carbohydrates, is a key sugar monomer. C-type lectins (CTLs) are a major family of animal lectins that bind in a Ca²⁺-dependent manner to mono- and oligosaccharides [36]. Sequence similarity analysis (Table 2) and multiple sequence alignment (Figure 2) revealed notable variation among the examined protein sequences, except for *H. diversicolor*. Dodenhof et al. [37] also observed high heterogeneity among perlucin amino acid sequences in their study of shell-forming nacre proteins. Phylogenetic analysis further showed that Hdh-Perlucin clusters with *H. diversicolor* in the molluscan clade, with the nearest members of this cluster being perlucin 8 and perlucin 6 isoforms.

In mollusks, the mantle is associated with the growth and coloration of the shell, both of which are critical for shell biomineralization [38,39]. Tissue-specific expression analysis identified the mantle (which showed the highest expression), hemocytes, gills, and digestive gland as key organs expressing *Hdh-Perlucin* in Pacific abalone. Rousseau et al. [40] reported that the mantle, gills, and hemolymph are the primary calcium compartments involved in shell biomineralization. Molluscan mantle tissue proteins can influence the formation of diverse structural layers within their shells [41]. The mantle-secreted perlucin is a shell nacre protein and has been reported to be responsible for the formation of the nacreous layer in molluscan shells. Other shell-forming nacre proteins in mollusks include carbonic anhydrase, Pif, perlustrin, AP7, perlin, lustrin A, BMSP, perlwapin, N19, N66, N14, MSI60, N40, AP24, and perlinhibin [42,43,44]. During the biomineralization process, the gills absorb Ca²⁺ ions from the external environment, which are then transported to the mantle epithelium via hemocytes and precipitated onto the organic matrix [45]. Hdh-Perlucin, like other perlucin proteins, binds Ca²⁺-dependent carbohydrates, which may play a role in crystal nucleation, as well as in the crystallization, morphology, and polymorphism of calcium carbonate during biomineralization [46]. The reason for the expression of *Hdh-Perlucin* in the digestive gland tissue remains unclear. However, transcriptome analysis of the windowpane oyster, Placuna placenta, identified six perlucin unigenes from the mantle tissue, one of which (c67461_g2) shows higher expression in the hepatopancreas [44].

The abalone shell begins to form during the transformation from larvae to juvenile [47]. In the early development stages of Pacific abalone, *Hdh-Perlucin* exhibits significant expression levels, particularly during the larval stages. The first shell (protoconch-I) develops during the trochophore larval stage, and the significantly higher expression of *Hdh-Perlucin* at this stage suggests that perlucin plays a role in the initial shell development (protoconch-I) of Pacific abalone. In blue mussels, perlucin is reportedly involved in the prodissoconch-I (PD1) shell formation, which corresponds to the PD1 in gastropod mollusks [48]. Additionally, larvae with silenced perlucin experienced reduced biomineralization, shell deformities, and even premature death in oysters [49].

Mollusks’ shell development continues steadily throughout their lifetime [50], encompassing both normal shell growth and the repair of damaged shells. Naturally, rapidly growing abalones exhibit faster shell development. The upregulated expression of *Hdh-Perlucin* in growth-specific expression analysis (slow vs. rapid growth) suggests that perlucin secretion is higher in faster-growing abalones, playing a vital role in rapid shell formation. The increasing trend in *Hdh-Perlucin* mRNA expression during the shell repair process further supports its strong affinity for calcium carbonate crystallization. A similar pattern was observed during the first two weeks of shell repair in the mussel *Hyriopsis cumingii* [29]. However, *Hdh-Perlucin* expression continued to increase after the second week, while *Hc-perlucin* expression decreased [29]. Although the exact reason for this difference is unclear, gene expression patterns can vary depending on the structural and functional properties of the gene and the species involved.

Abalones are susceptible to thermal stress, which can lead to reduced growth, survival, and shell development. The optimum temperature for Pacific abalone growth ranges from 18 °C to 22 °C [51]. Elevated temperatures can damage the shell microstructure by altering the expression of genes involved in shell biomineralization [50]. This study observed reduced *Hdh-Perlucin* mRNA expression in response to high temperatures (25 °C and 30 °C). Thermal stress may exacerbate acidification damage, further impairing shell growth by altering its microstructure [50]. Similar to the effects seen in the *Hdh-AP7* and *Hdh-AP24*, thermal stress significantly altered the expression profile of *Hdh-Perlucin*, potentially causing irregularities in the shell surface structure.

Starvation is a condition characterized by a significant deficiency in energy intake, falling below the level required to sustain an organism’s life. During the first two weeks of starvation, *Hdh-Perlucin* expression remained relatively unchanged, but it significantly decreased during the third and fourth weeks. However, a notable increase in *Hdh-Perlucin* expression was observed after prolonged starvation (during re-feeding), suggesting an increased shell deposition rate, possibly through the involvement of shell matrix proteins. Prolonged starvation depletes the body’s organic matter. Linard et al. [52] demonstrated that food concentration regulates shell growth in Pinctada margaritifera. High-fed oysters showed an abundance of transcripts for the shell nacre proteins shematrin 9 and Pif 177, likely due to increased energy transfer to the mantle tissue for shell formation [53]. Previous studies have shown that temperature and food availability are crucial factors influencing tissue and shell growth, primarily through their effects on shell matrix proteins [53]. During starvation, energy is derived from body reserves and tissue catabolism, leading to a reduction in organic matter when food is insufficient [54]. In larvae, starvation delays the time to reach metamorphosis and extends the overall duration of metamorphosis [55].

## 5. Conclusions

To our knowledge, this study is the first to isolate and characterize the full-length perlucin cDNA from Pacific abalone, while also analyzing the expression patterns of the perlucin protein across various tissues, embryonic and larval stages, growth phases, shell damage repair, prolonged starvation, and thermal stress. The presence of a Ca²⁺-dependent carbohydrate-recognizing CTLD and tissue-specific expression in biomineralizing compartments (mantle, gill, and hemocytes) suggest that *Hdh-Perlucin* functions as a shell biomineralizing nacre protein. Given its higher expression during early shell development, upregulation in different growth types of abalone, and increased expression during shell repair, perlucin can be considered a key biomarker gene for studying shell biomineralization. The observed downregulation during starvation and exposure to elevated temperatures provides new insights into the behavior of *Hdh-Perlucin* under nutritional and environmental stress. These findings may prove valuable for future studies on the structure–function relationships and physiological roles of *Hdh-Perlucin* in Pacific abalone and closely related mollusk species.

## Figures and Tables

**Figure 1 biology-13-00944-f001:**
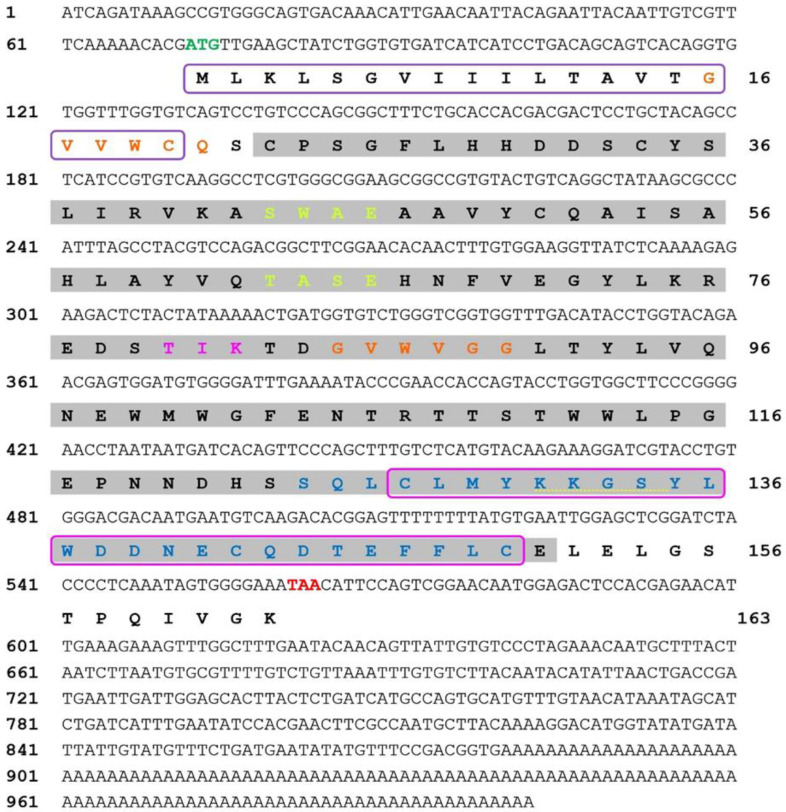
The complete nucleotide sequence and deduced amino acid residues of *Hdh-Perlucin* in Pacific abalone are shown. The numbers on the left and right of the sequences indicate the positions of the adjacent nucleotides and amino acid residues, respectively. The coding regions, which begin at the start codon and end at the stop codon, are displayed in bold green and red letters, respectively. A violet box highlights the signal peptide. Orange letters denote the N-myristoylation sites, while a shaded ash background in the coding region highlights the carbohydrate-binding C-type lectin domain (CTLD). Potential protein kinase C (PKC) phosphorylation sites are marked in pink, and casein kinase II (CK2) phosphorylation sites are highlighted in lime green. Yellow dotted underlines indicate cAMP- and cGMP-dependent protein kinase phosphorylation sites, and the CTLD signature is enclosed in a pink box. The blue letters signify the chitin-binding type-2 domain profile.

**Figure 2 biology-13-00944-f002:**
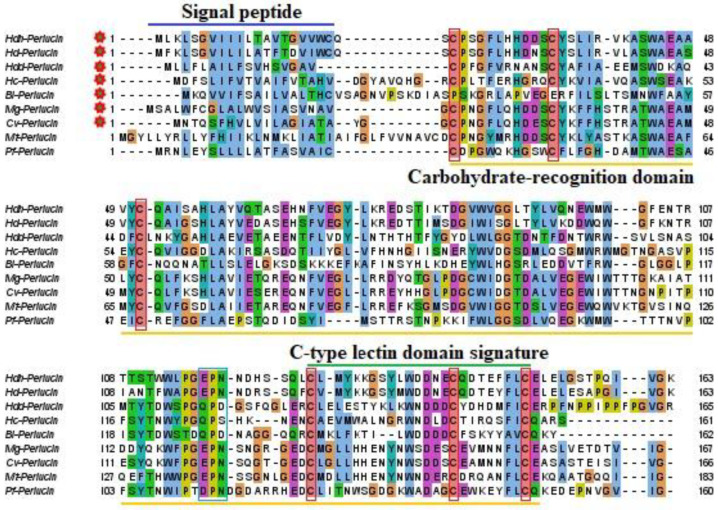
Multiple sequence alignment of perlucin protein sequence of invertebrates. The deep blue line is for the signal peptide. The orange line region signifies the carbohydrate-binding C-type lectin domain. The C-type lectin domain signature was denoted by a green line. The red box indicates the conserved cysteine residues. The blue box indicates the EPN motif.

**Figure 3 biology-13-00944-f003:**
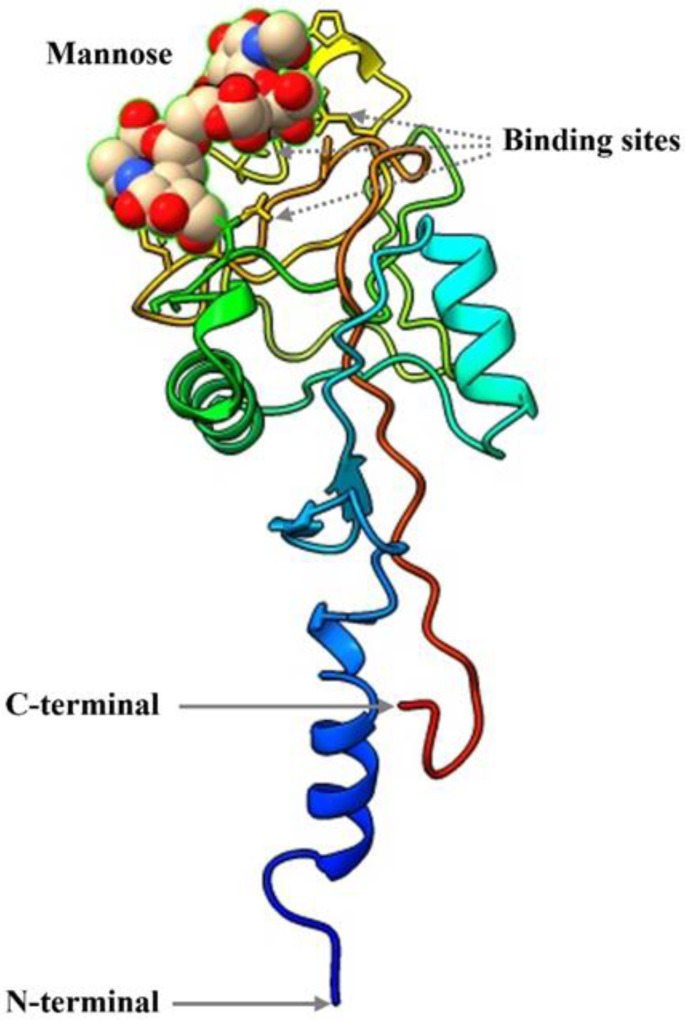
The Hdh-Perlucin tertiary structure includes a ligand, ligand binding site, and terminal region.

**Figure 4 biology-13-00944-f004:**
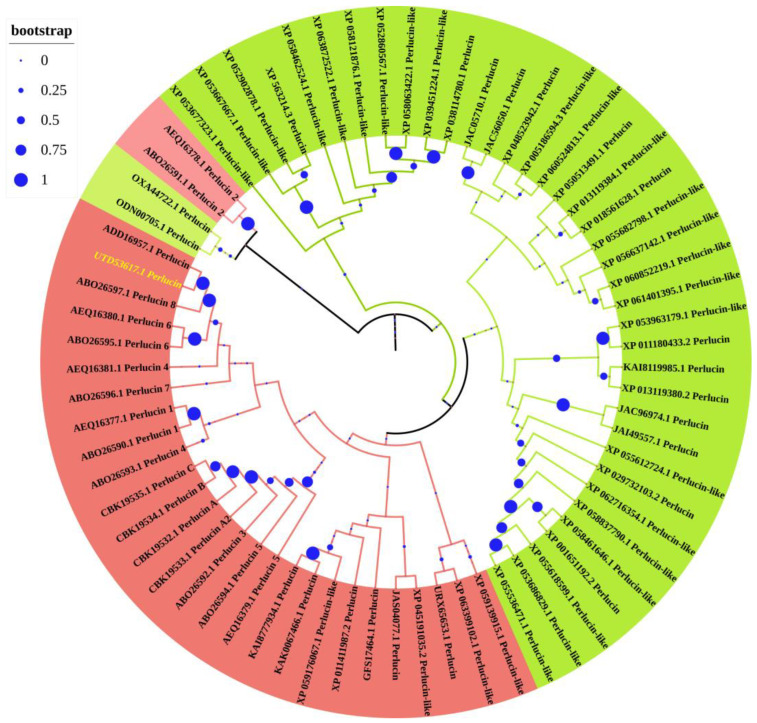
A protein sequence-based phylogenetic tree of perlucin was constructed using the maximum-likelihood method of different invertebrate species. The numbers at the nodes represent bootstrap probability. The lime-green cluster denotes arthropods, while salmon and light salmon are used to depict mollusks.

**Figure 5 biology-13-00944-f005:**
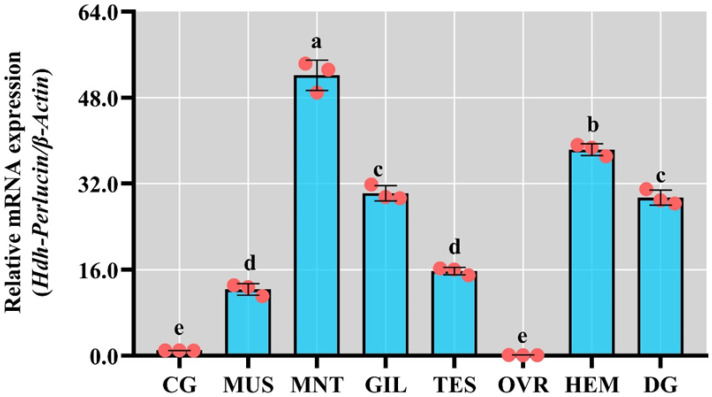
Relative expression of *Hdh-Perlucin* mRNA in different tissues of Pacific abalone. CG, cerebral ganglion; MUS, muscle; MNT, mantle; GIL, gill; TES, testis; OVR, ovary; HEM, hemocyte; DG, digestive gland. The lowercase letter above the bar indicates significant levels.

**Figure 6 biology-13-00944-f006:**
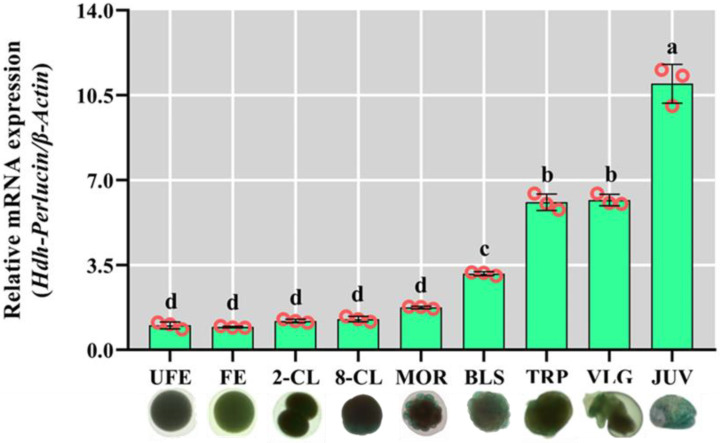
Developmental stage-specific relative mRNA expression of *Hdh-Perlucin* in Pacific abalone: UFE (unfertilized egg), FE (fertilized egg), 2-CL (2-cell), 8-CL (8-cell), MOR (morula), BLS (blastula), TRP (trochophore), VLG (veliger), JUV (juvenile). The lowercase letter above the bar indicates significant levels.

**Figure 7 biology-13-00944-f007:**
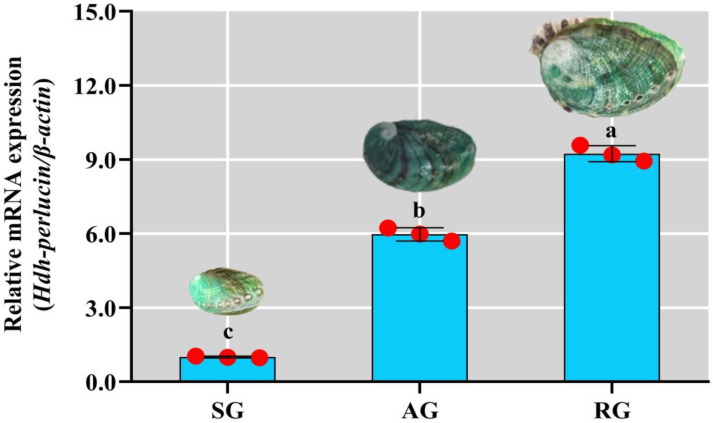
*Hdh-Perlucin* expression in mantle tissue of Pacific abalone exhibiting different growth patterns. SG, slow growth; AG, average growth; RG, rapid growth. The lowercase letter above the bar indicates significant levels.

**Figure 8 biology-13-00944-f008:**
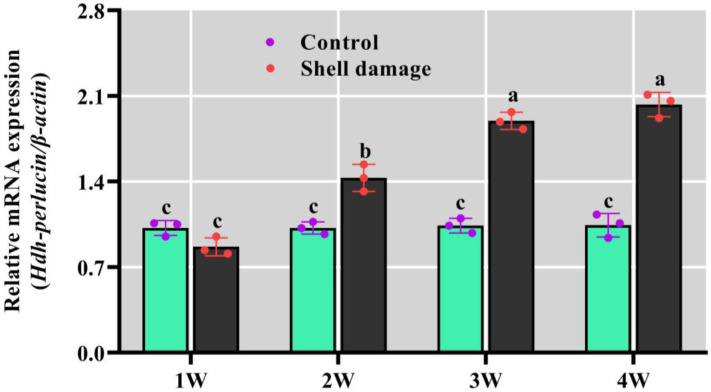
Relative mRNA expression of *Hdh-Perlucin* in Pacific abalone during injured-shell regeneration. The lowercase letter above the bar indicates significant levels. W, week.

**Figure 9 biology-13-00944-f009:**
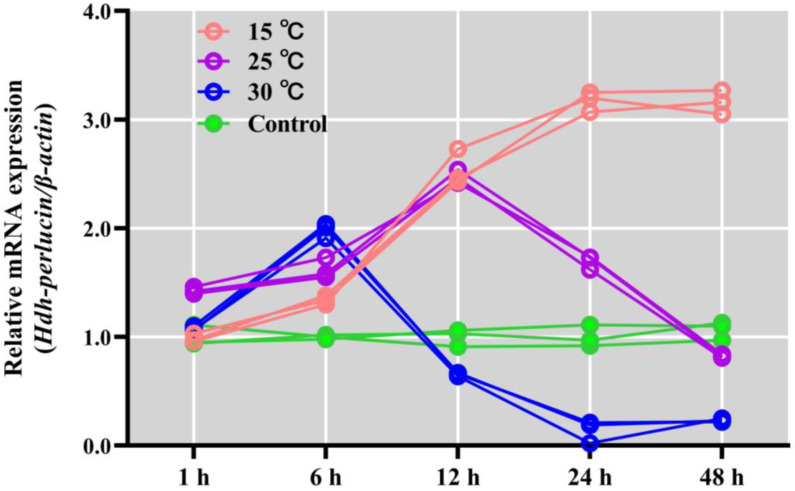
*Hdh-Perlucin* mRNA expression during Pacific abalone *H. discus hannai* thermal stress. A lowercase letter above the bar indicates significant levels.

**Figure 10 biology-13-00944-f010:**
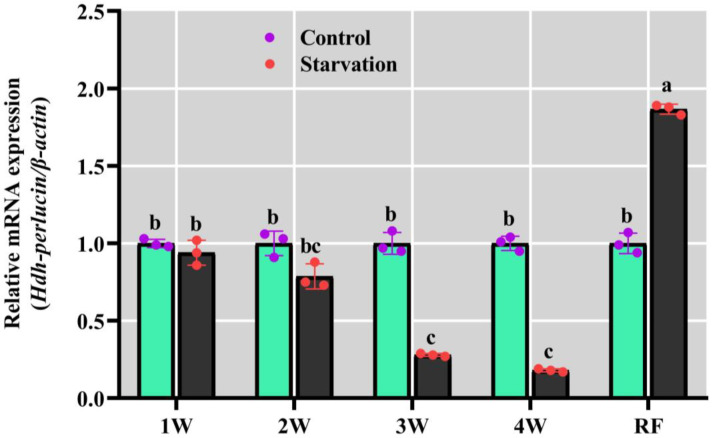
Expression pattern of *Hdh-Perlucin* in mantle tissue of starved Pacific abalone. The lowercase letter above the bar indicates significant levels. W, week; RF, re-feeding.

**Table 1 biology-13-00944-t001:** Primers used for cDNA synthesis, molecular cloning, and relative expression analysis.

Primer Name	Nucleotide Sequences	Purpose
Oligo dT	GGC CAC GCG TCG ACT AGT ACT TTT TTT TTT TTT TTT T	cDNA synthesis
OdT adapter	GGC CAC GCG TCG ACT AGT AC	
Perlucin-Fw	GAT CAT CAT CCT GAC AGC AG	RT-PCR
Perlucin-Rv	GAT CCA GAC AGC ATC AGT AC	
Perlucin-5R	GAT TAC GCC AAG CTT GTC AGT TCG CTT GCT CCC TTC ACG TG	RACE PCR
Perlucin-3R	GAT TAC GCC AAG CTT CAT CGC TCT ACC CTG CAA ACC ATC GG	
Hdh-Perlucin-qF	GAC ATA CCT GGT ACA GAA CG	qRT-PCR
Hdh-Perlucin-qR	GTA GAT CCG AGC TCC AAT TC	
Hdh-β-Actin-Fw	GAT AGT GCG AGA CAT CAA GG	
Hdh-β-Actin-Rv	GAG CTC GAA ACC TCT CAT TG	

**Table 2 biology-13-00944-t002:** Similarity percentage status of Hdh-Perlucin protein with different invertebrates.

Protein Name	Species		Accession No.	Protein Identity (%)
	Scientific Name	Common Name		
Hdh-Perlucin	*Haliotis discus hannai*	Pacific abalone	ABO26591.1	100
Hd-Perlucin	*Haliotis diversicolor*	Variously colored abalone	ADD16957.1	82.82
Mg-Perlucin	*Magallana gigas*	Pacific oyster	XP_011455487.3	45.39
Cv-Perlucin	*Crassostrea virginica*	Eastern oyster	XP_022330928.1	45.39
Mt-Perlucin	*Mytilus trossulus*	Pacific blue mussel	XP_063425845.1	45.39
Hdd-Perlucin	*Haliotis discus discus*	Disk abalone	ABO26590.1	39.26
Pf-Perlucin	*Pinctada fucata*	Akoya pearl oyster	JAS04076.1	37.5
Hc-Perlucin	*Hyriopsis cumingii*	Triangle shell mussel	AGI61062.1	36.02
Bl-Perlucin	*Bactrocera latifrons*	Solanum fruit fly	JAI49557.1	27.77

## Data Availability

The data included in this study can be found in the manuscript and Appendix A. The associated raw data’s are available upon request from the corresponding author.

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
