# Peer review of "The Isolation and Characterization of Perlucin in Pacific Abalone, Haliotis discus hannai: A Shell Morphogenic Protein with Potential Responses to Thermal Stress and Starvation"

_biology, 2024, doi:10.3390/biology13110944_

Round 1
Reviewer 1 Report
Comments and Suggestions for Authors
This study aimed to study the Isolation and analysis of a morphogenetic protein from abalone shells. The present study contributes application of a thought for future advanced molecular research in abalone and other mollusks. However, there are a few issues that need to be corrected before the paper can be published.
1. Title and Abstract. Please add the Latin for Pacific abalone, the object of study in the article.
2. Introduction. The background of the molluscan shell formation process could be appropriately streamlined, please add more research and applications of Perlucin on the remaining similar species, specifically.
3.Line 117. What is “LN2”, Is it liquid nitrogen?
4. Line 142. Is this the optimum temperature for abalone?
5. Line 158. Why was the thermal stress experiment conducted with one group at 15℃, shouldn't it have been higher than the control group?
6. Results 3.1. Please add the bp of the full length of the sequence. Add the labelling of Figure 1 to the text
7. Figure 4. The picture seems a little blurry and not so clear
8. Line 482. The article says the optimum temperature is 15-18℃, why is the water temperature in the breeding at 19-20℃, won't it be affected by heat stress?
9. Line 493. What are the criteria for determining abalone starvation, weight loss? Is there any data to support this. The nutrient deficiency caused by starvation, why not supply energy to the muscle first, but increase the expression of shell matrix protein, the author's explanation is not very clear, please explain!
10. There are some grammar mistakes in the MS. Please check the English writing carefully.
Author Response
Response to reviewer
This study aimed to study the Isolation and analysis of a morphogenetic protein from abalone shells. The present study contributes application of a thought for future advanced molecular research in abalone and other mollusks. However, there are a few issues that need to be corrected before the paper can be published.
- Title and Abstract. Please add the Latin for Pacific abalone, the object of study in the article.
Response: Recommended correction has been made.
- The background of the molluscan shell formation process could be appropriately streamlined, please add more research and applications of Perlucin on the remaining similar species, specifically.
Response: Molluscan shell formation process has been edited including other research work on closely related species.
- Line 117. What is “LN2”, Is it liquid nitrogen?
Response: Yes, we used commonly used abbreviated form of liquid nitrogen. However, full form of “liquid nitrogen” has been added during first time use.
- Line 142. Is this the optimum temperature for abalone?
Response: We apologize for misinformation due to a typical mistake. The optimum temperature for growth and survival of Pacific abalone is 18–22 °C (Lee et al., 2023). Correction has been made in the manuscript.
- Line 158. Why was the thermal stress experiment conducted with one group at 15℃, shouldn't it have been higher than the control group?
Response: Thank you for your comment. As the optimum temperature for Pacific abalone growth and survivability is 18–22 °C, our focus point was the expression pattern of Hdh-perlucin above and below the optimum temperature as well as response against maximum tolerance temperature. For this reason, we select 15 ℃ as below the optimum temperature, 20 ℃ as control (within optimum range), 25 ℃ as above the optimum temperature (within tolerate range) and 30 ℃ as the maximum tolerate temperature [acute temperature 31 ℃ (Xu et al., 2020)].
Xu, F., Gao, T. & Liu, X. Metabolomics Adaptation of Juvenile Pacific Abalone Haliotis discus hannai to Heat Stress. Sci Rep 10, 6353 (2020). https://doi.org/10.1038/s41598-020-63122-4
- Results 3.1. Please add the bp of the full length of the sequence. Add the labelling of Figure 1 to the text
Response: The full-length bp of perlucin cDNA sequence has been added in the result section. Figure 1 has been cited in the text.
- Figure 4. The picture seems a little blurry and not so clear
Response: Figure 4 has been changed as it seemed a little blurry.
- Line 482. The article says the optimum temperature is 15-18 ℃, why is the water temperature in the breeding at 19-20℃, won't it be affected by heat stress?
Response: Thank you for your pointful comment. We previously mentioned that an optimal temperature range of 15–18 °C was mistakenly presented for Pacific abalone. This temperature (likely 16–18 °C) is optimal for Australian blacklip abalone. However, the optimal temperature for Pacific abalone ranges from 18–22 °C. We have corrected this information in the manuscript. Based on this optimal temperature range, we maintained 19–20 °C for breeding Pacific abalone.
- Line 493. What are the criteria for determining abalone starvation, weight loss? Is there any data to support this. The nutrient deficiency caused by starvation, why not supply energy to the muscle first, but increase the expression of shell matrix protein, the author's explanation is not very clear, please explain!
Response: Thank you for your insightful comment. Yes, weight loss and body shrinkage were effects of starvation. During starvation, a weekly weight loss of 1.27 ± 0.31 g was observed. We provided sufficient food until the experiment began but did not supply energy to the muscle tissue. Our focus was on the expression pattern of Hdh-perlucin under energy starvation, as energy is essential for shell formation. Since food is the primary energy source, this led to a condition of food deprivation. In line 493, we explained that “a notable increase in Hdh-perlucin expression after prolonged starvation may suggest an increased shell deposition rate….” Here, “after prolonged starvation” refers to the period following the ‘completion of starvation during re-feeding’, not during the starvation period itself. Please refer to Figure 10 for clarification. I have added “during re-feeding” to the sentence for easier understanding.
- There are some grammar mistakes in the MS. Please check the English writing carefully.
Response: The grammatical mistakes have been carefully checked and corrected.
Reviewer 2 Report
Comments and Suggestions for Authors
This paper is well written and the assessment of perlucin gene expression is comprehensive. I only have a few minor comments.
1. It would be great if the authors could include a discussion on some other studies in Mollusca biomineralization genes, and discuss the expression pattern of perlucin in a broad evolutionary perspective. Consider citing the following papers:
1) Li, R., Zarate, D., Avila-Magaña, V., & Li, J. (2024). Comparative transcriptomics revealed parallel evolution and innovation of photosymbiosis molecular mechanisms in a marine bivalve. Proceedings B, 291(2023), 20232408.
2) Song, N., Li, J., Li, B., Pan, E., & Ma, Y. (2022). Transcriptome analysis of the bivalve Placuna placenta mantle reveals potential biomineralization-related genes. Scientific Reports, 12(1), 4743.
3) Song, X., Liu, Z., Wang, L., & Song, L. (2019). Recent advances of shell matrix proteins and cellular orchestration in marine molluscan shell biomineralization. Frontiers in Marine Science, 6, 41.
2. The justification for temperature treatments (15°C, 25°C, and 30°C) is missing.
Author Response
Response to reviewer
This paper is well written and the assessment of perlucin gene expression is comprehensive. I only have a few minor comments.
- It would be great if the authors could include a discussion on some other studies in Mollusca biomineralization genes and discuss the expression pattern of perlucin in a broad evolutionary perspective. Consider citing the following papers:
Response: Thank you for your comment. Some other studies in Molluscan biomineralization related genes have been included in the discussion part and discussed specially based on the recommended articles.
1) Li, R., Zarate, D., Avila-Magaña, V., & Li, J. (2024). Comparative transcriptomics revealed parallel evolution and innovation of photosymbiosis molecular mechanisms in a marine bivalve. Proceedings B, 291(2023), 20232408.
2) Song, N., Li, J., Li, B., Pan, E., & Ma, Y. (2022). Transcriptome analysis of the bivalve Placuna placenta mantle reveals potential biomineralization-related genes. Scientific Reports, 12(1), 4743.
3) Song, X., Liu, Z., Wang, L., & Song, L. (2019). Recent advances of shell matrix proteins and cellular orchestration in marine molluscan shell biomineralization. Frontiers in Marine Science, 6, 41.
- The justification for temperature treatments (15°C, 25°C, and 30°C) is missing.
Response: Thank you for your comment. As the optimum temperature for Pacific abalone growth and survivability is 18–22 °C, our focus point was the expression pattern of Hdh-perlucin above and below the optimum temperature as well as response against maximum tolerance temperature. For this reason, we selected 15 ℃ as below the optimum temperature, 20 ℃ as control (within optimum range), 25 ℃ as above the optimum temperature (within tolerate range) and 30 ℃ as the maximum tolerate temperature [acute temperature 31 ℃ (Xu et al., 2020)].
Xu, F., Gao, T. & Liu, X. Metabolomics Adaptation of Juvenile Pacific Abalone Haliotis discus hannai to Heat Stress. Sci Rep 10, 6353 (2020). https://doi.org/10.1038/s41598-020-63122-4.

Round 2
Reviewer 1 Report
Comments and Suggestions for Authors
The author has modified and improved the quality of the MS.